

# Investigating the lignocellulolytic gut microbiome of huhu grubs (*Prionoplus reticularis*) using defined diets and dietary switch

Jay Viswam[1], Mafalda Baptista[1,2], Charles K. Lee[1], Hugh Morgan[1] and Ian R. McDonald[1]

[1] Te Aka Mātuatua-School of Science, Te Whare Wānanga o Waikato-University of Waikato, Hamilton, Waikato, New Zealand
[2] Waikato Regional Council, Hamilton, Waikato, New Zealand

## ABSTRACT

The huhu beetle (*Prionoplus reticularis*) is the largest endemic beetle found throughout Aotearoa New Zealand, and is characterised by feeding on wood during its larval stage. It has been hypothesised that its gut microbiome plays a fundamental role in the degradation of wood. To explore this idea we examined the fungal and bacterial community composition of huhu grubs' frass, using amplicon sequencing. Grubs were reared on an exclusive diet of either a predominantly cellulose source (cotton) or lignocellulose source (pine) for 4 months; subsequently a diet switch was performed and the grubs were grown for another 4 months. The fungal community of cellulose-reared huhu grubs was abundant in potential cellulose degraders, contrasting with the community of lignocellulose-reared grubs, which showed abundant potential soft rot fungi, yeasts, and hemicellulose and cellulose degraders. Cellulose-reared grubs showed a less diverse fungal community, however, diet switch from cellulose to lignocellulose resulted in a change in community composition that showed grubs were still capable of utilising this substrate. Conversely, diet seemed to have a limited influence on huhu grub gut bacterial communities.

## INTRODUCTION

The huhu beetle (*Prionoplus reticularis*), a member of the long-horned beetle family (Coleoptera: Cerambycidae), is the largest endemic beetle found throughout Aotearoa New Zealand (*Edwards, 1959*; *Hosking, 1978*). The adult beetle does not eat, but the huhu grub larvae (Cerambycids) are xylophagous (feed on wood) and are thought to be dependent on the gut microbial community for lignocellulose degradation (*Reid et al., 2011*). The life cycle of the huhu beetle involves holometabolism (complete metamorphosis), and the larval stage can take two to three years (*Edwards, 1961*). Traditionally, 'huhu' is the Māori name for the larval stage (scarabaeiform grub-like), also known as *tunga haere* or *tunga rakau*; however, currently huhu is often used as the name for all life stages of the beetle

Corresponding author
Ian R. McDonald,
irmcdon@waikato.ac.nz

(*Viswam, 2016*). Generally, huhu grubs play a beneficial ecological role by assisting in decomposition and nutrient cycling. In nature throughout the larval stage huhu grubs solely feed on wood, leaving a trace of sheared wood and egested faecal frass (*Edwards, 1961*, *1959*; *Hosking, 1978*; *Viswam et al., 2018*). Huhu grub larvae are known to feed on a broad range of dead trees (*Edwards, 1959*; *Rogers, Lewthwaite & Dentener, 2002*), and this adaptability to various diets makes it an ideal candidate for studies to identify microorganisms potentially involved in lignocellulose degradation.

Previously, we demonstrated the suitability of rearing huhu grubs under laboratory conditions and identified variation in fungal communities between dietary groups using community fingerprinting techniques (*Viswam et al., 2018*). In this study we aim to understand how the huhu gut microbiome responds to a diet of cellulose (cotton) or lignocellulose (pine), and the switch between them, by analysing amplicon sequencing of marker genes. The ITS region of the rRNA operon was used to identify fungi and the V4–V3 region of the 16S rRNA gene was used to identify the bacteria, to provide taxonomic identification and relative abundance of both fungal and bacterial gut microbiomes (*Viswam, 2016*).

Generally, factors like pH, host specificity (individual nature of grubs and co-evolutionary effects), life stage, host environment and diet are known to influence the gut community structure (*Behar, Yuval & Jurkevitch, 2008*; *Mohr & Tebbe, 2006*; *Santo Domingo et al., 1998*; *Schmitt-Wagner et al., 2003*; *Yun et al., 2014*). Evidence from previous studies suggests that the crucial factors that shape the gut community in insects are diet and taxonomy (*Colman, Toolson & Takacs-Vesbach, 2012*). For this reason we aimed at minimising the effects of environmental variables (including temperature, water availability, and food source) by rearing grubs in a controlled environment (*Viswam et al., 2018*). The microbiomes of a number of other xylophagous insects have been studied, most notably wood eating termites and cockroaches (*Berlanga, 2015*; *Berlanga & Guerrero, 2016*; *Brune, 2014*; *Ni & Tokuda, 2013*). The bacterial community in these insects has been found to be dominated by Spirochaetes, Bacteroidetes, Firmicutes and Proteobacteria (*Boucias et al., 2013*; *Mikaelyan et al., 2015*).

Lignocellulose, the main component of plant cell walls, is a complex biopolymer composed of the polysaccharides cellulose (20–50%) and hemicellulose (15–35%) and lignin (18–35%). The degradation of lignocellulose requires the synergistic action of cellulases, hemicellulases, lignases, and lytic polysaccharide monooxygenases (*Obeng et al., 2017*) to deconstruct the complex lignocellulosic structure, and therefore presents a challenging food source. Cellulose, a major component of plant material, is a linear polysaccharide of glucose units, is degraded by cellulases, and therefore is a more accessible food source. Using lignocellulose or cellulose as the sole diet available to huhu grubs provides an opportunity to identify which members of the gut microbiome play a role in degradation of these significant components of plant cell walls, the only diet of these xylophagous insects.

Based on the assumption that huhu grubs rely on their gut microbiome for lignocellulose degradation, we hypothesise the heterogeneous lignocellulose (pine) diet and the homogeneous cellulose (cotton) diet will lead to distinct microbial communities.

Furthermore, we anticipate that 1) the removal of lignocellulose (pine) from the diet will cause a decrease in diversity (*i.e.*, loss of specialist fungi potentially involved in breaking down lignin and hemicellulose) and 2) the re-introduction of lignocellulose in the diet will restore those specialists in the gut microbiome, at least to some extent.

## MATERIALS AND METHODS

### Sample site and collection of huhu grubs

Grubs used in this study were collected from a single Kahikatea tree stump in Claudelands Reserve, Hamilton, New Zealand. Only healthy, average sized (3–5 cm) huhu grubs were collected. The number of huhu grubs used in this study was limited by restrictions in the permit that was required for collections. Individual grubs were grown in isolation on either a sterile cellulose (cotton) or a sterile lignocellulose (pine) diet as previously described (*Viswam et al., 2018*). Frass was collected aseptically daily, sieved to separate the frass from wood or cotton debris, and stored at −20 °C until required.

Four grubs were initially grown on the cellulose (cotton) diet and another four grubs were grown on the lignocellulose (pine) diet. Weight gain and other physical characteristics of each grub were monitored to ensure optimal growth conditions were maintained. Huhu grubs were reared on the defined diets for 4 months and frass was collected aseptically and stored at −20 °C before diets were switched. Grubs initially grown on cotton were switched to pine, and those grown on pine were switched to cotton (Fig. 1). Grubs were then closely monitored to ensure that no adverse physical effects were induced by the dietary switch. All grubs were then grown on the switched diets for 4 months and the frass was collected as before.

### DNA extraction, PCR and sequencing

Frass collected during the third and fourth month after transfer to a new diet was used for DNA extraction (as previously described in *Viswam et al., 2018*). Fungal community was studied using the ITS region, which was PCR amplified using adapted primers (IDT, New Zealand), specifically ITS1F (5′-CCATCTCATCCCTGCGTGTCTCCGACTCAG-unique IonXpress barcode-GATCTTGGTCATTTAGAGGAAGTAA-3′) (*Gardes & Bruns, 1993*) and the reverse primer ITS 2 with label sequence (5′-CACTACGCCTCCGCTTTCCTCT CTATGGGCAGTCGGTGAT-GCTGCGTTCTTCATCGATGC-3′) (*White et al., 1990*). The bacterial community was studied using the V4 region of the 16S rRNA gene, which was amplified using the adapted primer set 515F (5′-CCATCTCATCCCTGCGTGTCT CCGACTCAG-unique IonXpress barcode-GATGTGCCAGCMGCCGCGGTAA-3′) and 806R (5′-CCACTACGCCTCCGCTTTCCTCTCTATGGGCAGTCGGTGATGGACT ACHVGGGTWTCTAAT-3′). Each reaction contained 1x PCR buffer (Life Technologies, New Zealand), 0.2 mM dNTPs (Roche Diagnostics, New Zealand), 0.02 mg/mL Bovine serum albumin (BSA), 0.5 µM of each primer, 1 U Platinum Taq (Life Technologies, Carlsbad, CA, USA), 0.5 µM $MgCl_2$, 1 ng of DNA, and the reaction was made up to 25 µL with milli-Q $H_2O$. PCR reactions were completed for each extraction in triplicate to reduce stochastic PCR bias (*Ihrmark et al., 2012*; *Polz & Cavanaugh, 1998*). Thermal cycling conditions were: 94 °C for 3 min, then 30 cycles of

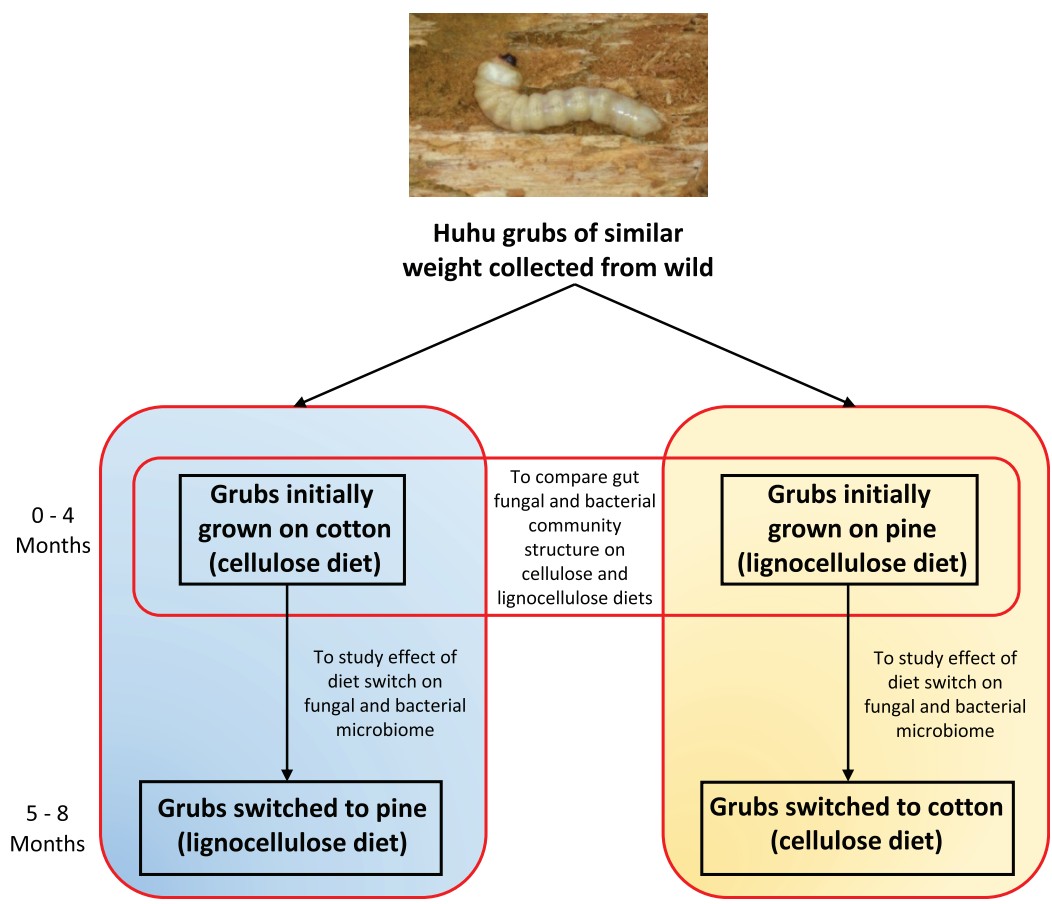

**Figure 1 Outline of the experimental plan.** Healthy, average sized (3–5 cm) huhu grubs were collected in the wild from a Kahikatea log. Grubs were then reared individually on either the cotton or pine diets for 4 months, diets were then switched and grubs grown for a further 4 months on the new diet. Frass was collected aseptically daily, sieved to separate the frass from wood or cotton debris, and stored at −20 °C until required. Frass collected during the third and fourth month on each diet was used for DNA extraction to study the gut microbiome (Photo credit: Jay Viswam).

94 °C for 45 s, 58 °C for 1 min (55 °C for bacterial primers), 72 °C for 90 s, and a final extension of 72 °C for 10 min. PCR amplicons were size verified on a 1% SYBR safe stained agarose gel and were quantified using Qubit dsDNA HS assay kit and Qubit 2.0 Fluorometer (Life Technologies, Carlsbad, CA, USA) (Data were collected as previously described in *Viswam, 2016*). Negative extraction and PCR controls were cleaned and analysed along with the samples. Each sample was diluted to 26 pM and mixed together equally to make an equimolar library. Sequencing was performed using Ion PGM sequencing 400 Kit, Ion 318 chip and Ion Personal Genome Machine System (ThermoFisher Scientific, Waltham, MA, USA) at the DNA sequencing facility, University of Waikato.

## Sequence analysis

Sequencing primers and indexing barcodes were removed with cutadapt (v2.3) (*Martin, 2011*). Sequencing reads were processed using DADA2 (v1.14.1) to generate amplicon
variants (ASVs) (*Callahan et al., 2016*). For taxonomic affiliations of fungal ITS sequences, the representative ASV sequences were classified using the UNITE reference database (*Abarenkov et al., 2010*). Taxonomy of bacterial 16S rRNA gene sequences was assigned to each variant using the SILVA database (v138) (*Quast et al., 2013*). ASVs classified as mitochondria, chloroplast, eukaryotes were removed from the dataset. This resulted in a bacterial data set with a median number of 23,255 sequences (ranging from 31,654 to 6,974), and a total of 469 ASVs, and a fungal data set with a median number of 6,201 sequences (ranging from 33,779 to 2,105), and a total of 178 ASVs. Plotting of rarefaction curves (vegan v2.6.4) showed that the expected species richness was likely to have been captured (Fig. S1).

Sequences and metadata were combined with phyloseq (v1.34.0) (*McMurdie & Holmes, 2013*). Compositional bar plots and principal component analysis (PCA) were created with microViz (v0.11.1) and used to visualise community composition and structure between initial and switched diet groups. For compositional bar plots data was transformed in proportions of total counts per sample, and for PCA data was transformed with the centre log ratio, upon which the aitchison distance was calculated. A permutational analysis of variance (PERMANOVA) was used to check for differences in beta diversity between diets, using the vegan package implementation in microViz, with 999 permutations. The multivariate homogeneity of group dispersions (variances) was checked with betadisper, in vegan, followed by pairwise comparisons of group mean dispersions using permutest, also in vegan, with 999 permutations. Alpha diversity estimates were performed by subsampling 100 times, with replacement, using phyloseq. Kruskal-Wallis tests were performed to compare the response of different diets, as implemented in package ggpubr. Analyses were performed in R 4.0.5 (*R Core Team, 2024*).

## RESULTS

### Were there differences in the initial structure of the communities (alpha and beta diversity) on the two different diets (cotton *vs.* pine)

Fungal communities were dominated by either Ascomycota or Basidiomycota, respectively, in two cotton-reared grubs each (Fig. 2A). At the genus level it was interesting to see that three out of four grubs reared on cotton displayed high dominance (>90% relative abundance) of one single genus, namely *Stachybotrys* (Ascomycota) or *Apiotrichum* (Basidiomycota) (Fig. 2B). The grub that did not display an overwhelming abundance of one single genus showed high relative abundance of ASVs assigned to Ascomycota, and ASVs that could not be assigned further than kingdom Fungi, suggesting that if those ASVs had been assigned to a lower rank the outcome could have been similar to the other three grubs. Overall, the results suggest that during the 4 months of cotton rearing the dominant ASVs in the frass were representative of one single genus per individual.

Fungal communities of pine-reared grubs were dominated by phylum Ascomycota (Fig. 2C), and at the genus level displayed higher diversity than cotton-reared grubs (Fig. 2D). Genera *Scheffersomyces*, *Scytalidium*, *Penicillium* and *Parafabraea* (all

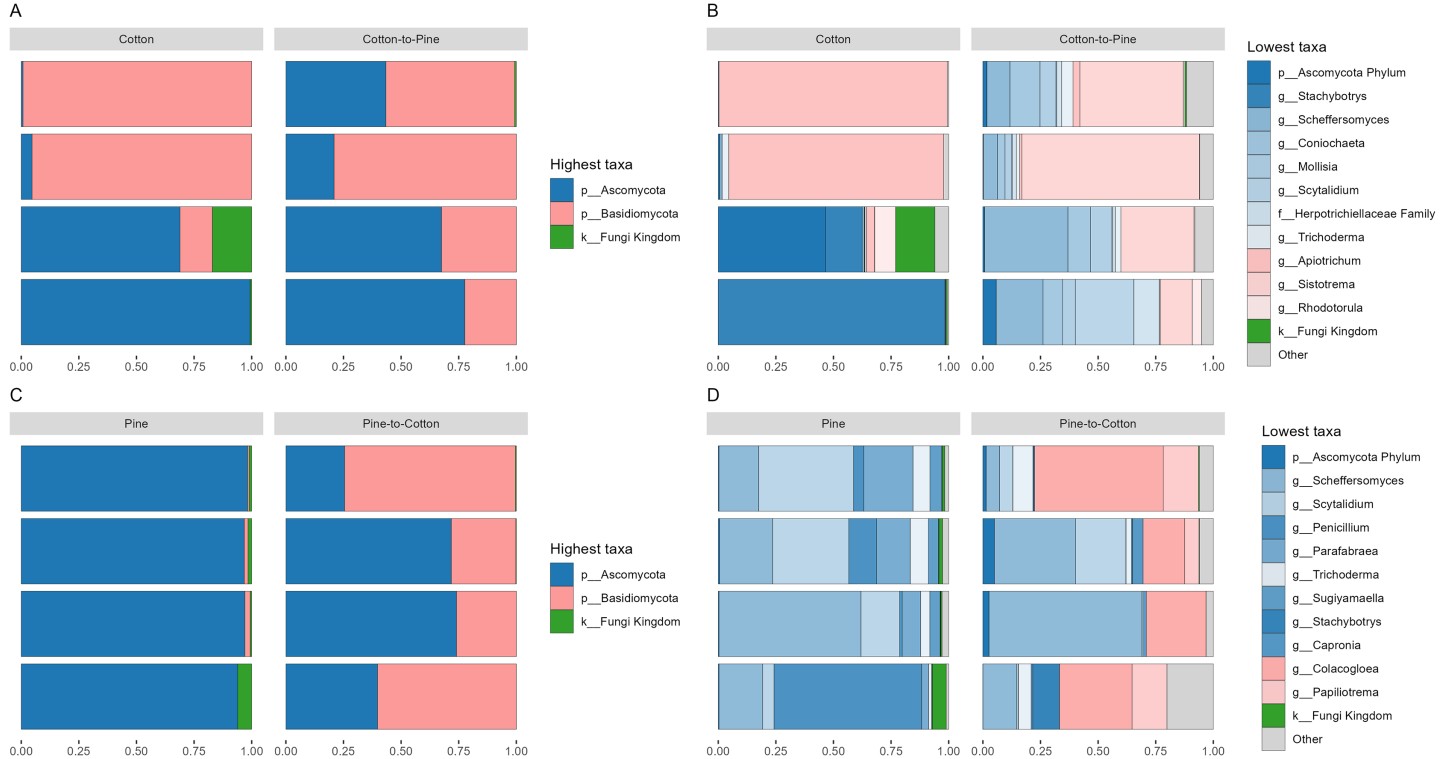

**Figure 2 Compositional barplots of the fungal community in the four different diets.** (A and B) Show cotton and cotton to pine diets, for the highest and lowest rank that could be assigned, respectively. (C and D) Show pine and pine to cotton diets for the highest and lowest rank that could be assigned, respectively. Each horizontal bar plot corresponds to the same individual grub assessed after 4 months on the initial diet and after 4 months on the diet switch. Abundance is shown as proportions of total reads. Top abundant ASVs (accounting for >75% of total counts) are shown, the remaining ASVs are grouped as "Other". In (B and D) the lowest level taxa are coloured in shades of the colour of the highest level taxa to which they belong.

Ascomycota), accounted for more than 75% of the relative abundance in all four grubs. In terms of richness (number of observed ASVs) a difference was seen between different diets (Fig. 3A), albeit not significant at the $p < 0.05$ level, with the pine diet having a higher richness than the cotton diet, which can also be seen at the genus level (Figs. 2B and 2D). A Principle coordinate analysis (PCA) indicated a clear separation between fungal communities of grubs reared on cotton or pine diets (Fig. 3B), and PERMANOVA showed that the four diets could be separated (F-statistic = 3.04, $p$-value = 0.001). A pairwise comparison showed the pine-to-cotton diet to be different from the other diets ($p < 0.05$), and a PERMANOVA performed on the cotton or pine diets separately showed that cotton *vs.* cotton-to-pine and pine *vs.* pine-to-cotton diets could be separated (Fig. S2).

Bacterial communities in grubs raised initially on a cotton diet were dominated by phyla Bacteroidota, Proteobacteria (Pseudomonadota in the 2021 revision by the International Committee on Systematics of Prokaryotes, but referred to here by the assignment by SILVA v138) and Acidobacteriota. The phylum Verrucomicrobiota was also amongst the most relatively abundant in some grubs (Fig. 4A). On the cotton diet when looking at the most abundant ASVs, the response of the four grubs was not similar. A higher variability

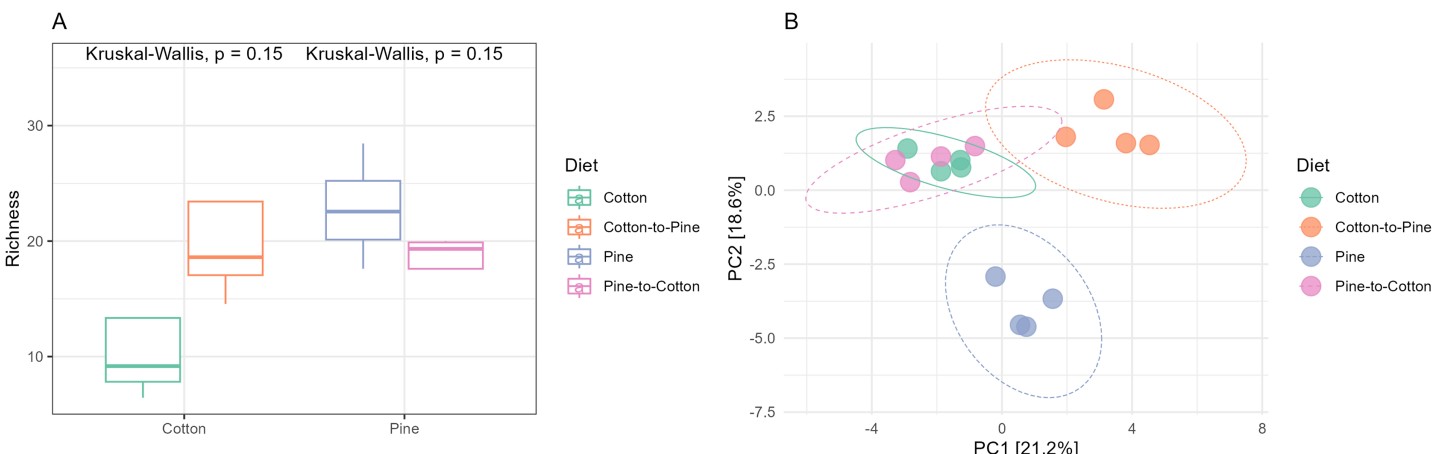

**Figure 3 Comparison of the fungal community structures in the four different diets.** (A) Alpha diversity (observed richness) for each diet. Differences in alpha diversity were evaluated using the Kruskal-Wallis test. (B) Beta diversity for each diet. A PCA was performed on the aitchison distances of a centred log ratio transformed ASVs data set. Ellipses are overlaid at the 0.95 level. A PERMANOVA showed that the four diets could be separated (F-statistic = 3.04, *p*-value = 0.001).

could be seen, shown by the fact that for two grubs the most abundant ASVs could not account for more than 50 to 60% of relative abundance (Fig. 4B).

Bacterial communities in grubs raised initially on a pine diet were dominated by Acidobacteriota, Proteobacteria and Actinobacteriota (Fig. 4C), and all four grubs showed similar results in terms of the most abundant ASVs. In the pine diet 12 ASVs accounted for more than 75% of the relative abundance (Fig. 4D). In terms of alpha diversity, the pine diet had a higher number of observed ASVs ($p < 0.05$ level), then the pine-to-cotton diet (Fig. 5A). A Principle coordinate analysis (PCA) indicated a clear separation between bacterial communities of grubs reared on cotton or pine diets (Fig. 5B), and PERMANOVA showed that the four diets could be separated (F-statistic = 3.41, *p*-value = 0.001). A pairwise comparison showed the pine-to-cotton diet to be different from the cotton and cotton-to-pine diets ($p < 0.05$), and a PERMANOVA performed on the cotton or pine diets separately showed that cotton *vs.* cotton-to-pine and pine *vs.* pine-to-cotton diets could be separated (Fig. S3).

## After the diet switch, were there changes in communities within individual hosts away from the initial structure (cotton *vs.* cotton-to-pine, and pine *vs.* pine-to-cotton)

Before diet switch fungal communities were usually dominated by a single phylum. However, after diet switch all grubs displayed noticeable relative abundances of both Ascomycota and Basidiomycota (Figs. 2A and 2C). Grubs switched from cotton to pine showed a much higher diversity of abundant fungal genera, than when being reared initially on cotton (Fig. 2B). Genera *Scheffersomyces* and *Scytalidium* (which were also abundant in grubs initially reared on pine, Fig. 2D) showed high relative abundance, alongside genera *Sistotrema* (Basidiomycota) and *Coniochaeta* (Ascomycota) which did not show high abundance on the other diets (Fig. 2B).

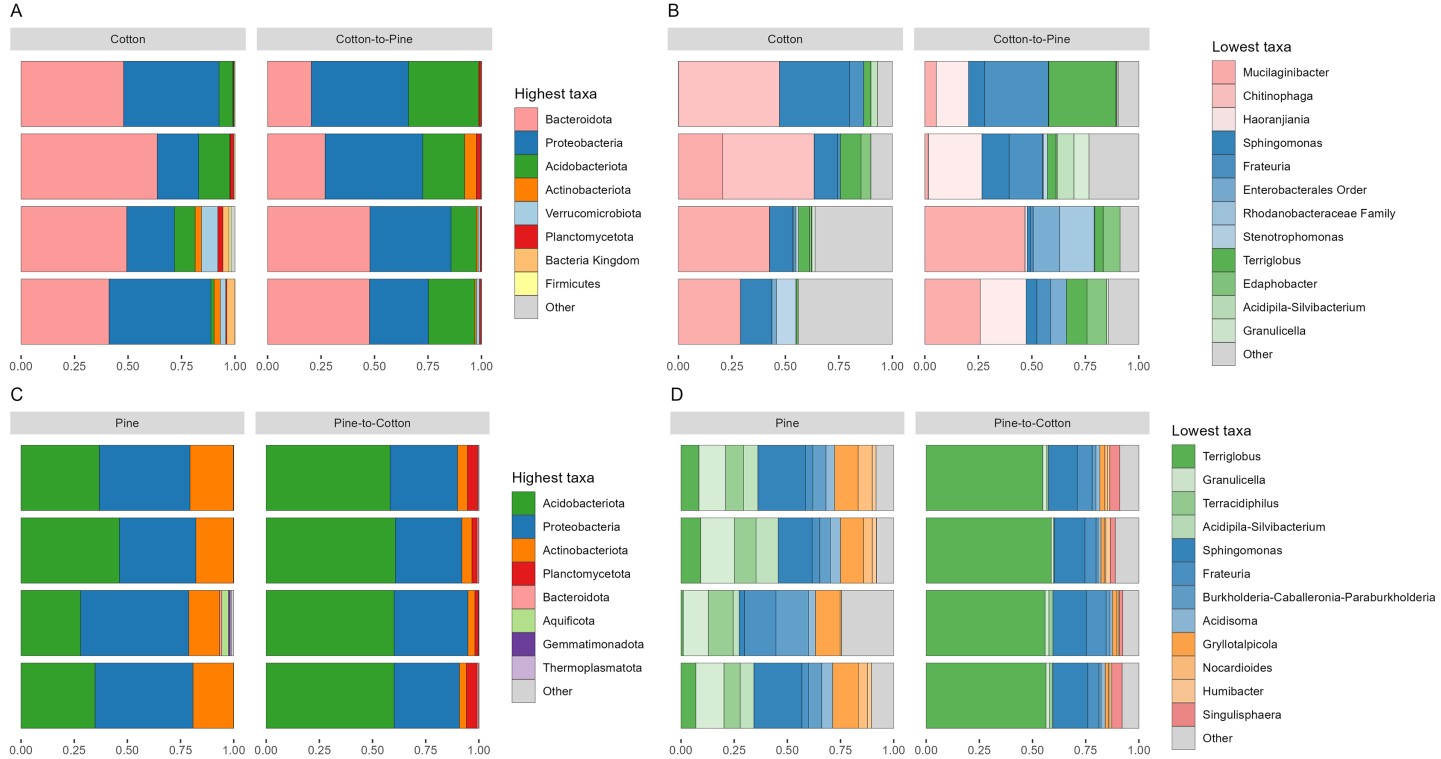

**Figure 4 Compositional barplots of the bacterial community in the four different diets.** (A and B) Show cotton and cotton to pine diets, for the highest and lowest rank that could be assigned, respectively. (C and D) Show pine and pine to cotton diets for the highest and lowest rank that could be assigned, respectively. Each horizontal bar plot corresponds to the same individual grub assessed after 4 months on the initial diet and after 4 months on the diet switch. Abundance is shown as proportions of total reads. Top abundant ASVs (accounting for at least >50% of total counts) are shown, for easier visualisation the remaining ASVs are grouped as "Other". In (B and D) the lowest level taxa are coloured in shades of the colour of the highest level taxa to which they belong.

In grubs switched from pine to cotton the genera *Scheffersomyces* and *Scytalidium* (which were also abundant in pine reared grubs, Fig. 2D) continued to show high relative abundance, alongside genera *Colacogloea* (Basidiomycota) and *Papiliotrema* (Basidiomycota) which did not show high abundance on the other diets (Fig. 2D). Interestingly, genera *Penicillium* and *Parafabraea* (which were abundant in pine reared grubs) did not show high relative abundances in the pine to cotton diet (Fig. 2D). In terms of alpha diversity or richness (number of observed ASVs) no difference could be seen between different diets (Fig. S2B).

While principle coordinate analysis (PCA) indicated a clear separation between fungal communities of grubs reared initially on cotton or pine diets (Fig. 3B), grubs switched from cotton to pine showed little similarity with pine reared grubs, but grubs switched from pine to cotton clustered together with the cotton reared grubs, suggesting that the community structure responded to the last 4 months of rearing on a cotton diet. PCA was also undertaken to compare two diets instead of all four, however Permanova analyses did not indicate any patterns not seen in the four diet analysis (Fig. S2).

Bacterial communities after diet switch showed some change at the phylum level, with the switch from cotton to pine (Fig. 4A) resulting in a decrease in abundance of

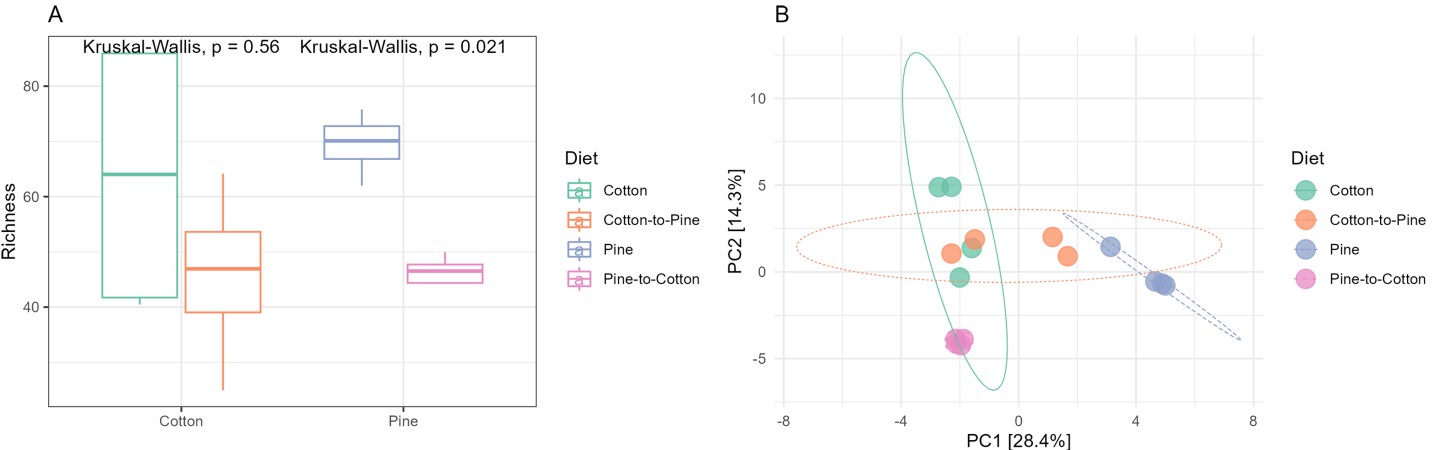

**Figure 5 Comparison of the bacterial community structures in the four different diets.** (A) Alpha diversity (observed richness) for each diet. Differences in alpha diversity were evaluated using the Kruskal-Wallis test. (B) Beta diversity for each diet. A PCA was performed on the aitchison distances of a centred log ratio transformed ASVs data set. Ellipses are overlaid at the 0.95 level. A PERMANOVA showed that the four diets could be separated (F-statistic = 3.41, *p*-value = 0.001).

Bacteriodota and an increase in abundance of Acidobacteriota and Proteobacteria. At the genus level (Fig. 4B) ASVs that did not show high relative abundance in the cotton diet, but showed high relative abundance in the cotton to pine diet included *Haoranjiania*, Enterobacterales that could only be assigned to order, and Rhodanobacteraceae that could only be assigned to family.

The switch from pine to cotton (Fig. 4C) resulted in an increase in the abundance of the phylum Acidobacteriota, and a decrease in Proteobacteria and Actinobacteriota. While at the genus level (Fig. 4D) two ASVs, *Terriglobus* and *Sphingomonas*, accounted for 75% of the relative abundance, compared to the 12 ASVs seen on the pine diet. In terms of alpha diversity or richness (number of observed ASVs) there appears to be a reduction in diversity upon switching of diets, though these were not significant as shown by Kruskal-Wallis tests (Fig. 5A). A PCA indicated a clear separation between bacterial communities of grubs reared on cotton and those switched from cotton to pine (Fig. 5B), that was also noticeable in the compositional barplots (Fig. 4). Grubs reared on pine and switched from pine to cotton showed completely separate communities (Fig. 5B). PCA was also undertaken to compare two diets instead of all four, however Permanova analyses did not indicate any patterns not seen in the four diet analysis (Fig. S3).

## Did these changes occur in the direction of the initial structure of the communities of the diet to which they were switched (pine-to-cotton *vs.* cotton and cotton-to-pine *vs.* pine)

If we compare the structures of the fungal or bacterial communities of grubs growing on cotton or pine initially, or at the end of the study, using the diversity barplots (Figs. 2 and 4) or PCA (Figs. 3B and 5B) only one diet switch showed a change in the direction of the initial structure. This was seen in the PCA of fungal communities of grubs growing on cotton after switching from growth on pine, which had communities similar to grubs

initially grown on cotton (Fig. 3B). However for bacterial communities change in the direction of the initial community structures was not clearly seen.

## DISCUSSION

In this study we provided huhu grubs with a sterile non-degraded pine wood substrate. In the wild, wood-feeding insects either feed on previously degraded wood or utilize symbionts to aid in extracting essential nutrients (*Geib et al., 2008*). Therefore, a key concern in this study was that grubs would remain healthy when reared on non-degraded wood. The grubs showed continued growth and health (*Viswam et al., 2018*), which was taken as an indication of their ability to feed on non-degraded pine wood.

The changes in the huhu gut fungal community when grubs were switched between diets supports the idea that diet is a significant driver for the maintenance of microorganisms responsible for lignocellulose (pine) degradation. Fungi are known to be involved in lignocellulose (pine) degradation, both in nature and within the insect gut (*Brune, 2014*; *Geib et al., 2008*; *Gibson & Hunter, 2010*; *Hyodo et al., 2000*, *2003*).

For grubs grown initially on lignocellulose (pine) an almost complete dominance of Ascomycota could be seen. This could be due to their ability to survive under stressful conditions (*Blanchette et al., 2004*; *Worrall & Wang, 1991*). Furthermore, the resistance to enzymatic degradation of lignocellulose (pine) may have resulted in suppression of growth of fungi that cannot play a role in lignocellulose degradation (*Brimner & Boland, 2003*; *Fokkema, 1973*; *Francis & Read, 1994*; *Schirmbock et al., 1994*; *Whipps, 1987*).

The frass of lignocellulose (pine) reared grubs was dominated by *Scheffersomyces*, *Scytalidium*, *Penicillium*, and *Parafabraea*. *Scheffersomyces* are known to produce plant cell wall degrading enzymes (*Martiniano et al., 2013*; *Scordia et al., 2012*; *Suh et al., 2003*; *Tanimura et al., 2015*). *Scytalidium* is a soft rot fungus which thrives in conditions unfavourable to white and brown rot fungi (*Blanchette et al., 2004*), and the use of water-logged wood (*Viswam et al., 2018*) provided ideal conditions for soft rot wood decay. Soft-rot fungi preferentially utilise cellulose and hemicellulose (*Hamed, 2013*), consistent with limited lignin modification previously seen in frass of huhu grubs (*Reid et al., 2011*). However, soft rot fungi have been shown to degrade lignocellulose in the wood feeding Asian longhorned beetle, *Anoplophora glabripennis* (*Geib et al., 2008*). *Penicillium spinulosum*, a saprotrophic fungi which degrades dead organic matter, was previously identified as the most abundant taxa in the lignocellulose reared gut fungal community of huhu grubs (*Williams, 2011*). The presence of *Sugiyamaella* as a less abundant component of the gut mycobiome was consistent with its presence in wood and frass of lignicolous beetles (*Houseknecht et al., 2011*; *Kurtzman & Robnett, 2007*). Diet switch from lignocellulose to cellulose (pine to cotton) resulted in a change in the relative abundances of the gut fungal community, to being dominated by four genera, two of which were not abundant on the pine diet, *Colacogloea* (Basidiomycota) and *Stachybotrys* (Ascomycota), but maintaining *Scheffersomyces* and *Scytalidium* (both Ascomycota) with high relative abundances in pine to cotton reared grubs.

For the fungal population observed in grubs grown initially on cellulose (cotton) the most relative abundant fungal genera were *Apiotrichum* and *Stachybotrys* which are

known to be present in environments with high cellulose content, and with cellulose degrading capabilities (*Gujjari et al., 2011*; *Middelhoven, Scorzetti & Fell, 2001*; *Murtoniemi, Nevalainen & Hirvonen, 2003*; *Rytioja et al., 2014*; *Stevens & Payne, 1977*; *Suh et al., 2003*; *Wang et al., 2015*). After switching diet from cellulose to lignocellulose (cotton to pine) *Sistotrema*, *Scytalidium* and *Scheffersomyces* displayed the highest relative abundances, suggesting that these genera are involved in lignocellulose degradation. *Scytalidium* and *Scheffersomyces* were also abundant in grubs initially raised on lignocellulose, but the presence of *Sistotrema*, a brown rot fungus and wood degrader (*Son et al., 2010*), as the dominant fungus indicates a change in diet has selected for fungi potentially involved in lignocellulose degradation.

Grubs initially grown on lignocellulose had an abundance of soft rot fungi (*Scytalidium*), whereas grubs switched to lignocellulose were abundant in a well-known wood degrading Basidiomycete (*Sistotrema*). This suggests that the huhu grub gut fungal microbiome is capable of using different wood rot mechanisms (white-rot, brown-rot, or soft-rot), or a combination of mechanisms dependent on diet and environment.

Previous studies have shown that diet is important in structuring the gut bacterial community, particularly in insects that feed on lignocellulose-derived substrates (*Colman, Toolson & Takacs-Vesbach, 2012*). In xylophagous termites the gut communities are similar, but different to detritivorous termites, whose gut communities are more similar to other detritivorous Coleoptera and Diptera. Potentially suggesting the presence of a microbiome necessary for lignocellulose metabolism in xylophagous termites (*Breznak & Brune, 1994*).

In this study bacterial diversity was low, consistent with other studies of lab-reared insects (*Geib et al., 2009*), and in line with the bacterial community composition of huhu grubs in the wild (*Reid et al., 2011*). The absence of Firmicutes as an abundant phyla was unexpected, as previous studies of insect gut diversity reported Firmicutes to be dominant (*Yun et al., 2014*). Firmicutes were also dominant and metabolically active in the gut of wild huhu grubs (*Reid et al., 2011*), and were involved in lignocellulose degradation in cockroaches (*Bertino-Grimaldi et al., 2013*). However, since the low abundance of Firmicutes was consistent in all grubs in this study, and there was adequate sequencing coverage, it was assumed that this phylum was at low abundance in the gut of all eight individuals collected in this study.

The frass of lignocellulose (pine) reared grubs was dominated by Proteobacteria and Acidobacteriota, with the most abundant genera being *Sphingomonas* and *Frateuria*, and *Terriglobus*, respectively. *Sphingomonas* were abundant in all grubs regardless of diet, and have previously been identified in the guts of nematodes (*Dirksen et al., 2016*), Chinese white pine beetle (*Hu et al., 2014*), cotton bollworm (*Priya et al., 2012*) and spiders (*Zhang et al., 2018*), while *Frateuria* has previously been isolated from the insect *Hyalesthes obsoletus* (*Lidor et al., 2019*). The Acidobacteriota genus *Terriglobus* was present in all diets, and has been reported to be capable of degrading xylan, a hemicellulase found in plant cell walls (*Eichorst, Kuske & Schmidt, 2011*). Diet switch from lignocellulose to cellulose (pine to cotton) resulted in an increase in the relative abundance of *Terriglobus*, leading to speculation that these bacteria are involved in plant polymer degradation. Actinobacteriota

showed high relative abundance in grubs grown on lignocellulose (pine) but a lower relative abundance in grubs switched to cellulose (pine to cotton). Actinobacteriota are commonly present in insect guts (*Geib et al., 2009*; *Kaltenpoth, 2009*; *Lefebvre et al., 2009*), and play a role in hemicellulose degradation in cerambycids (*Park et al., 2007*). In several studies the presence of Actinobacteriota was suggested to play a role as defensive mutualists (*Scott et al., 2008*; *Zucchi, Prado & Consoli, 2012*) providing protection against pathogenic bacteria (*Kaltenpoth, 2009*).

For the bacterial population observed in grubs grown initially on cellulose (cotton) a prevalence of Bacteroidota could be seen. This is consistent with their ability to contribute to cello-oligomer degradation in the cellulose-hydrolysing process (*Zhang et al., 2014*). Specifically, the genera *Mucilaginibacter* has been shown to degrade xylan present in wood (*Khan et al., 2013a*, *2013b*; *Oh et al., 2016*). In those grubs switched to lignocellulose (cotton to pine) Bacteroidota continue to be a prevalent phylum, with *Haoronjiania* (*Zhang et al., 2016*) displaying high relative abundances. Overall, Proteobacteria and Acidobacteriota were abundant regardless of diet, and only slight changes in the bacterial community structure at the genus level after dietary switch suggests that the bacterial community in the grub's gut might not have been related to providing a means for utilisation of cellulose and lignocellulose.

Recently the horizontal transfer of plant cell wall degrading enzyme genes from bacteria and fungi into the genomes of Coleoptera (beetles), and the endogenous expression of these enzymes, has been shown to be widespread (*McKenna et al., 2019*; *Tokuda, 2019*). The genome of the huhu beetle (*Prionoplus reticularis*) has not yet been sequenced, but it would be insightful to examine the genome when it becomes available, however the presence of plant cell wall degrading enzymes in the genome of beetles does not exclude the role of the microbiome in the degradation of plant material by the insect.

In conclusion, our hypothesis that diet would directly impact the gut microbiome was supported, with the gut fungal community of lignocellulose-reared grubs being different from that of cellulose-reared grubs. Soft rot fungi capable of producing cell wall degrading enzymes were identified, which could be potentially involved in making polymer carbohydrate available from lignocellulose. It was also hypothesised that removal of lignocellulose from the diet would cause a decrease in diversity. In fact, in terms of relative abundance, cellulose-reared grubs showed extremely low numbers of abundant fungal ASVs, a situation contrasting with a higher diversity in lignocellulose-reared grubs, even when these were switched to a cellulose diet. The re-introduction of lignocellulose in the diet showed that grubs were still capable of utilising this substrate, albeit grubs initially grown on lignocellulose had an abundance of soft rot fungi, whereas grubs switched from cellulose to lignocellulose were abundant in a well-known wood degrading Basidiomycete. In contrast, bacterial communities shared reasonable similarity regardless of diet, suggesting that bacteria play a lesser role in the metabolism of lignocellulose and cellulose by the huhu grub microbiome. The small sample size in this study prevented the inclusion of a non-diet switching control group, which could have precluded any changes caused by non-dietary factors (*e.g.*, developmental changes over time). However, we believe that the changes seen in this study are most likely to have been due to changes in the diet.

## ACKNOWLEDGEMENTS

We especially thank Roanna Richards-Babbage (Waikato University) for assistance with DNA extraction and sample preparation for sequencing that was carried out at the Waikato DNA Sequencing Facility, Hamilton, New Zealand.

### Funding

The authors received no funding for this work.

### Competing Interests

Ian R. McDonald is an Academic Editor for PeerJ.

### Author Contributions

- Jay Viswam conceived and designed the experiments, performed the experiments, analyzed the data, prepared figures and/or tables, authored or reviewed drafts of the article, and approved the final draft.
- Mafalda Baptista analyzed the data, prepared figures and/or tables, authored or reviewed drafts of the article, and approved the final draft.
- Charles K. Lee conceived and designed the experiments, authored or reviewed drafts of the article, and approved the final draft.
- Hugh Morgan conceived and designed the experiments, authored or reviewed drafts of the article, and approved the final draft.
- Ian R. McDonald conceived and designed the experiments, prepared figures and/or tables, authored or reviewed drafts of the article, and approved the final draft.

### Data Availability

The DNA sequences are available at SRA: PRJNA1048488.

### Supplemental Information

Supplemental information for this article can be found online at http://dx.doi.org/10.7717/peerj.17597#supplemental-information.

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
