# Peer review of "Investigating the lignocellulolytic gut microbiome of huhu grubs (Prionoplus reticularis) using defined diets and dietary switch"

_PeerJ, doi:10.7717/peerj.17597_

## Round 0.1 · original submission · Major Revisions

Your manuscript has now been assessed by two reviewers and myself. We all agree that this is a well-written manuscript with some potentially interesting findings.

The reviewers have identified several areas where the manuscript could be improved. In particular I agree that the introduction needs more ecological context - both of the focal species and a comparison to other insects with similar dietary niche. Most importantly, all the analyses presented in this manuscript are at the group/treatment level, and seem not to make use of the ability to measure within-individual changes following dietary switch - we can't assume they all have the same rate of change across diets, for example. This is especially important as your design did not include a no-switch control to allow you to separate treatment effects from drift over time.

I look forward to seeing a revised version of the manuscript.

·

Basic reporting

No comment

Experimental design

No comment

Validity of the findings

No comment

Additional comments

Paper summary:
This paper takes wuhu grubs from a population in the wild and rears them in a laboratory environment on two different diets consisting of cellulose (cotton) or lignocellulose (pine). After four months, each group of grubs is switched onto the different diet for another four months. At the end of each dietary period, fracas from the grubs were taken for ITS and 16S rRNA sequencing to describe the fungal and bacterial components of the microbiome respectively. The authors then interpret changes in the alpha diversity and community composition, arguing that the mycobiome is much more responsive to changes in diet.

General comments:
Overall, this study includes an elegant experimental design which allows for pairwise comparisons of the same individuals from each dietary regime. Despite the limited sample size, which is acknowledged by the authors, the results are still informative in combining measures of both fungal and bacterial diversity. However, the study could be improved in several ways as described below:
- The experimental design utilises a before and after comparison, whereby all individuals undergo a dietary switch. This means that individuals are compared against themselves at an earlier time-point, rather than against a control population reared on a single diet type and sampled at the same time. As the population was originally collected from the wild and then transferred to the lab, it is possible that some of the longitudinal changes observed in the microbiome and currently attributed to the effect of diet are actually being driven by a wider microbiome shift in response to captivity (see Mays et al 2021 https://doi.org/10.1093/femsle/fnab121 for an example of such shifts in insects). It would also be informative to know more about the tree that the grubs were originally collected from to know what there starting diet (and therefore possibly microbiome) was likely to be in the wild.

- The introduction and discussion are currently written with a species-specific focus and make little reference to the rich literature studying the microbiome of other xylophagous insects (most notably wood-eating cockroaches and termites). It would improve the paper to broaden the perspective and include more discussion comparing wuhu grubs with what is already known about the bacterial and fungal microbiome of other xylophagous insects (see Brune 2014 - https://doi.org/10.1038/nrmicro3182 - and Ni and Tokuda 2013 - https://doi.org/10.1016/j.biotechadv.2013.04.005 - as introductions to some of this work on termites.

- Currently the methods are unclear about whether each grub are reared in isolation or not. Presumably this is the case to allow each fracas sample to be assigned to a specific individual, but it should be made more explicit.

- The introduction and methods are missing information on the ecological context of diet switching in wild wuhu grubs. How often and on what timescales would grubs be expected to switch diet in the wild, and how does this match with the four-month dietary periods used in the experiment? How well do the two different diets provided in the experiment represent the variability in diets that wild wuhu grubs are likely to experience.

- The colours used in Figures 2 and 4 could be improved to allow for clearer comparison between plots A and B. In plot A, each highest-level taxa could be given a different colour (i.e., red), and then lowest-level taxa in plot B that belong to the same highest-level taxa could be coloured with shades of that same colour (i.e., different reds, pinks, oranges). This would make it easier for the reader to see the same higher-level groupings of plot A within the lower classification of plot B.

- The discussion could be more cautious when speculating about the possible functional roles of the different fungal and bacterial taxa identified. Without a more direct assessment of functional attributes (i.e., use of shotgun metagenomics), link to function based on 16s or ITS data alone will only be correlational and should be treated as such (see Jovel et al 2016 https://doi.org/10.3389/fmicb.2016.00459 and Antwis, Harrison and Cox’s Microbiomes of Soils, Plants and Animals: An Integrated Approach (Ecological Reviews) for discussion of approaches for understanding microbiome function).

Specific comments:
Alongside the general comments mentioned above, there are a few comments that could be addressed at specific points of the manuscript.
Line 73 – Phylogeny rather than taxonomy.
Line 284 – Geib et al 2008 reference does not provide data on bacterial diversity for which it is used to support. Claim that bacterial diversity of insects is low is not true in all cases, e.g., termites.
Line 328 – plant cell wall degrading enzymes

Reviewer 2 ·

Basic reporting

The manuscript is generally well written with very few grammatical mistakes and only a few suggestions to rephrase certain sentences to improve clarity. The structure of the manuscript conforms to standard and the references are complete and well-cited. The raw sequence data is clearly signposted and accessible. The figures are relevant and of good quality, with a few minor suggestions for improvement, although the figure legends need more detail. The introduction would benefit from some additional detail on the differences in the structure of lignocellulose and cellulose and the different metabolic requirements for the digestion of these two diet components, to make it clear how the hypotheses were formed. The introduction would also benefit from adding more detail on the relevance of the dietary shifts to the ecology of the huhu grubs in their natural environment.

Line 60: expand on the differences previously detected in fungal communities among dietary groups
Line 167: clarify ‘one single genus’, per individual? As there were two dominant genera found in the group as a whole
Line 183: remove ‘with’ after alongside
Line 201: Grammar – change ‘relative abundance’ to ‘relatively abundant’
Line 220: it is unclear what is meant by ‘two by two’, I suggest rephrasing for clarity
Line 231: Be more specific, what about the fungal community supports the idea
Line 238: Clarify what is meant by the ‘recalcitrant nature’ of lignocellulose
Line 8273: Unclear phrasing of ‘potentiated fungi potentially involved’
Figure 1: needs a proper legend with more detail describing the experiment. It might also be useful to show the timeline from collection of the grubs to the end of the experiment.
Figures 2 and 4: The ‘other’ category seems to have lines separating the taxa within this group, I recommend removing these lines to improve visual clarity.
Figures 3 and 5: It would be interesting to visually show the changes within individuals after the diet switch, for instance by adding arrows to the ordination joining samples from the same individual before and after the switch.
Figure S1: the legend needs more detail to explain what the vertical and horizontal lines represent.
Figure S2: the y-axes need to specify what the observed number is of (ASVs?). Also it would be interesting to see the changes within individuals occurring after diet switch by joining samples from the same individual with lines.

Experimental design

This is an original piece of primary research within the scope of the journal. The research questions are well defined, but as stated above, further detail in the introduction would help to contextualise the experiment and how it fills a knowledge gap. The authors have relied heavily on citing a previous study and consequently the methods are currently missing some key details. More information is needed on how the diets were prepared (e.g. were they sterile?), how the grubs were selected for the study, how they were housed (individually or in experimental groups?) and the sampling regime. Some of these details are alluded to throughout the manuscript, but need to be explicitly stated in the methods. It is of concern that no experimental control groups were included in the study design, therefore one is unable to compare the diet switches to a ‘no switch’ group. This would have allowed the authors to control for changes that might have occurred over time (e.g. stochastic or developmental) but not due to the diet switch.

Validity of the findings

The authors have been cautious of not over overstating their findings given the limited sample size. However, I have serious reservations that in its current form, the authors cannot properly answer the question they set out to. The current analyses (PERMANOVA and Kruskal-Wallis tests) compare all four diet categories at once as ‘like for like’, which does not help to ask the specific questions the authors are trying to answer, and is potentially obscuring some interesting patterns. It also ignores the fact that repeat samples within individuals were taken. I suggest restructuring the results to ask each question in turn as follows: A) were there differences in the initial structure of the communities (alpha and beta diversity) on the two different diets (cotton vs. pine). B) After the diet switch, were there changes in these communities within individual hosts away from this initial structure (cotton vs cotton-to-pine and pine vs. pine-to-cotton. C) Did these changes occur in the direction of the initial structure of the communities of the diet to which they were switched (pine-to-cotton vs cotton and cotton-to-pine vs. pine).
The full results of the analyses performed should be reported in the main text and not just in the figures or figure legend.
Lines 280 and 320: the authors describe a lack of change or a ‘lesser effect’ of diet on the bacterial vs fungal communities. I suggest rephrasing slightly as it is difficult to make these assertions without more rigorous statistical analyses.

Additional comments

The authors have furthered scientific knowledge of the effects of dietary change on the gut microbial communities of a lesser studied, non-model host species, the huhu grub, which is a valuable contribution to the field. Furthermore, consideration of both the bacterial and fungal components of the community is still rare and as such is appreciated. Overall, the manuscript is well written and the figures are well-presented, with only a few minor suggestions for improvement. However, a major concern is that the current structure of the results and analyses does not allow for robust testing of the hypotheses outlined. I hope the suggestions made above will help to make the most of the data that is available.

---

## Round 0.2 · Minor Revisions

Your manuscript has now been assessed by the two original reviewers and myself, and we all find the paper much improved. The only remaining query I would like to see addressed was raised by Reviewer 2, who rightly points out that at the moment there is no formal test in the manuscript of two of your key hypotheses.

As you treat diet as a 4 level factor, your Kruskal-Wallis (alpha diversity) and PERMANOVA (beta diversity) tests are simply testing for mean/centroid differences. But 'these diets have different mean trait values' is not as informative as pairwise tests of group differences linked to your hypotheses.

I also agree with reviewer 2 that figure S2 should be in the main text.

Once you have made these changes the work will not need to go back to review.

·

Basic reporting

The paper is well written with appropriate references throughout. I’m pleased to see the authors have taken the time to respond to each of the points raised in the previous round of peer review, and believe that the added section to the introduction on the differences between the two diets helps provide the additional context necessary to understand the study fully. Furthermore, the change in colours used in Figures 2 and 4 has helped make it easier to interpret them and link the plots labelled A and C with plots B and D respectively.

Experimental design

The study involves a clear experimental design with a clear series of research questions being addressed. The inclusion of Figure 1 is especially helpful in providing a broad-level overview of the experimental design and is valuable for assisting the reader in understanding each stage of the results that follow. The new additions to the method section have helped improve clarity regarding the experimental design, resolving previous confusion about whether the huhu grubs were housed individually or not.

I appreciate the constraints placed on the authors regarding the sample size available for this study and how this prevented them from including a non-switching control group. It would be helpful to acknowledge this within the paper with a statement that considers if the microbiome changes could have been caused by other non-dietary factors, e.g. developmental changes over time (even if diet is the more likely explanation).

Validity of the findings

The manuscript has been improved with the reworked results section that provides a clearer narrative structure and presents answers to each research question in turn. The analyses are appropriate for the questions being addressed.

It would be useful to provide the underlying data on body size and growth that is used to determine that grubs are growing healthily (this could be in the supplementary material rather than the main text, but should be available somewhere).

Additional comments

None

Reviewer 2 ·

Basic reporting

Thank you to the authors for providing the additional detail and clarity requested. I only have some additional suggestions on figures pertaining to the main hypotheses/aims.
I suggest moving Fig. S2 to the main text as it is very relevant to the hypotheses outlined. In addition, some lines joining the boxplots to indicate the group-level shifts occurring (cotton with cotton-to-pine / pine with pine-to-cotton) would aid the reader in understanding and interpretation. If figure space is limited, one solution could be to combine figures 2/3 and 4/5 together, perhaps with simplifying the barplots to one bar per group instead of one per individual. The individual-level variation in composition shown in current figs 2 and 4 could then be moved into the supplementary, as the author’s have stated this is not the main focus of the paper (this is just a suggestion for if moving figure S2 to the main text is problematic).

Similarly, I appreciate the author’s consideration of my suggestion of adding lines showing the individual trajectories of grubs’ microbiomes in the pca plots. I don’t think it is necessary to have the extra figures in the supplementary, however the lines do help to see the group-level shifts from one diet to another. I would ask the authors to consider adding some further indication of these group-level shifts (cotton and cotton-to-pine / pine and pine-to-cotton) to figs. 3 and 5.

Experimental design

I appreciate the extra detail provided in the methods. However, there remains some ambiguity as to the sample collection timeline and what the datapoints therefore represent. The authors state that ‘Frass was collected aseptically daily, sieved to separate the frass from wood or cotton debris, and stored at -20 oC until required’ and ‘Frass collected during the third and fourth month after transfer to a new diet was used for DNA extraction’. Does this mean that that the one datapoint per individual grub per diet represents a combination of the frass collected daily over the third and fourth months sequenced together as one sample, or that only the frass collected at a single timepoint at the end of each diet phase was used for sequencing? Some clarification here would be beneficial.

Mention of the statistical analyses performed for testing differences in alpha diversity among groups is need in the methods (i.e. Kruskal Wallis tests).

Validity of the findings

I appreciate the restructuring of the results section for ease of reading, and accept that the authors have limited sample size (and therefore power) to investigate individual-level patterns. However, my concerns on whether the current analyses are sufficient to answer the author’s hypotheses remain. The authors state their main hypotheses as follows ‘1) the removal of lignocellulose (pine) from the diet will cause a decrease in diversity (i.e., loss of specialist fungi potentially involved in breaking down lignin and hemicellulose) and 2) the re-introduction of lignocellulose in the diet will restore those specialists in the gut microbiome, at least to some extent’. These hypotheses require testing for the difference in microbiomes (alpha/beta diversity) between the following groups; for hypothesis 1) pine vs pine-to-cotton, and for hypothesis 2) cotton vs cotton-to-pine. I currently do not see these analyses presented in the results. Rather than running separate tests or separating out the diets in different figures as the author’s have done in response to my previous comments, I suggest a simple solution would be to keep the PERMANOVA and Kruskal-Wallis tests of all 4 diet groups, but include some post-hoc tests and report these in the relevant results sections. This would allow the authors to say not only that the 4 diet groups differed overall (as the reporting is currently limited to), but which groups differed from one another (cotton v pine, cotton vs cotton-to-pine etc.) and therefore allow them to address the hypotheses more completely.

---

## Round 0.3 · accepted · Accept

Congratulations on the acceptance